# Quantum Diffusion Models for Few-Shot Learning

## Abstract

Modern quantum machine learning (QML) methods involve the variational optimization of parameterized quantum circuits on training datasets, followed by predictions on testing datasets. Most state-of-the-art QML algorithms currently lack practical advantages due to their limited learning capabilities, especially in few-shot learning tasks. In this work, we propose three new frameworks employing quantum diffusion model (QDM) as a solution for the few-shot learning: label-guided generation inference (LGGI); label-guided denoising inference (LGDI); and label-guided noise addition inference (LGNAI). Experimental results demonstrate that our proposed algorithms significantly outperform existing methods.

## 1 Introduction

Quantum machine learning (QML) has emerged as a powerful tool for automated decision-making across diverse fields such as finance, healthcare, and drug discovery[1–4]. However, in the realm of few-shot learning, where only a limited amount of data is available for training, QML demonstrates suboptimal performance. In classical machine learning, diffusion models have been validated as effective zero-shot classifiers and hold significant potential for addressing few-shot learning problems[5, 6]. Nevertheless, in the domain of QML, the utilization of quantum diffusion models (QDMs) for few-shot learning remains largely unexplored[7]. This is primarily due to the limitations of quantum computing resources and the inherent noise associated with quantum computers, despite the QDM's demonstrated success in generative tasks[8].

In this work, we propose three new algorithms based on the QDM to address the few-shot learning problem. Our contributions are as follows:

- The QDM has demonstrated strong performance in generative tasks. Building on QDM's generative capabilities, we propose the **Label-Guided Generation Inference (LGGI)** algorithm to address the few-shot learning problem. Additionally, we introduce two algorithms: **Label-Guided Noise Addition Inference (LGNAI)** and **Label-Guided Denoising Inference (LGDI)**, to perform test inference respectively in diffusion and denoising stages.
- We compare our algorithms with other baselines in experiments on different datasets, which verified the superior performance of our proposed approaches.
- We conduct a comprehensive ablation study to evaluate the impact of various components and hyperparameters on the performance of the proposed algorithms.

## 2 Background

**Quantum Neural Network (QNN).** A Quantum Neural Network (QNN) has been used to perform various machine learning tasks. It typically consists of a data encoder $E(x)$ that embeds a classical

Submitted to the Second Workshop on Machine Learning with New Compute Paradigms at NeurIPS (MLNCP 2024). Do not distribute.

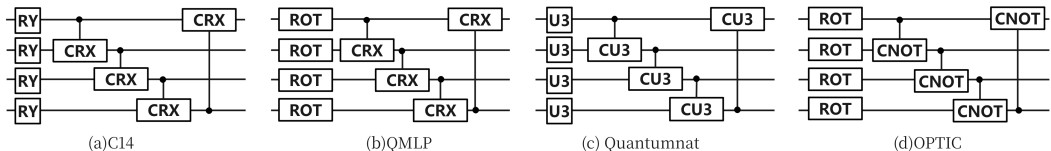

**Figure 1:** Various types of variational quantum circuits (VQC).

input $x$ into a quantum state $|x\rangle$, a variational quantum circuit (VQC) $Q$ that generates the output state, and a measurement layer $M$ that maps the output quantum state to a classical vector. Fig. 1 shows some VQC ansatz examples[9–12] used for QNNs. Given a training dataset, the input data $x$ is transformed into a quantum input feature map using $E(x)$. A parameterized VQC ansatz is then utilized to manipulate the quantum input feature through unitary transformations. Finally, the predicted classification is obtained by measuring the quantum state. The loss function is predefined to calculate the difference between the output of the QNN and the true target value $y$. Training a QNN involves iteratively searching for the optimal parameters in the VQC through a hybrid quantum-classical optimization procedure.

**Quantum Few-shot Learning (QFSL).** Few-shot learning (FSL) is a machine learning approach designed to address supervised learning challenges with a very limited number of training samples. Specifically, it involves a support set and a query set. The support set consists of a small number of labeled examples from which the model learns, encompassing $n$ classes, each with $k$ samples, hence called $n$-way $k$-shot learning. The query set is a collection of unlabeled examples that the model needs to classify into one of the $n$ classes. Existing solutions to the QFSL problem can be categorized into data-based, model-based, and algorithm-based methods[13]. Quantum Few-shot learning (QFSL) involves using QNNs as classifiers to solve QFSL problems[14, 15]. However, traditional algorithms used in QFSL often underperform due to the limited computational resources available and the noise present in real quantum devices.

**Quantum Diffusion Model (QDM).** Diffusion model (DM)[16, 17] is a popular approach for generating images and other high-dimensional data. It comprises two main processes: the diffusion process and the denoising process. During the diffusion process, noise is gradually added to the data over a series of steps, transforming it into a simpler distribution, as formulated by (1), in which $\mathcal{N}(\cdot; \mu, \Sigma)$ denotes the normal distribution of mean $\mu$ and covariance $\Sigma$, $\beta_t$ is a small positive constant that controls the amount of noise added at step $t$, and $\mathbf{I}$ is the identity matrix.

$$q(x_t|x_{t-1}) = \mathcal{N}(x_t; \sqrt{1-\beta_t}x_{t-1}, \beta_t\mathbf{I}) \tag{1}$$

The denoising process aims to learn how to reverse the forward process and incrementally remove noise to generate new data from the noise, with its training objective formulated by

$$\mathbb{E}_{q(x_{0:T})}\left[\sum_{t=1}^{T} D_{\mathrm{KL}}\big(q(x_{t-1}|x_t, x_0)\|p_\theta(x_{t-1}|x_t)\big)\right], \tag{2}$$

in which $q(x_{t-1}|x_t, x_0)$ is the posterior distribution of the forward process and the parameterized model $p_\theta(x_{t-1}|x_t)$ can predict the data point at the previous step given the current noisy data point. The denoising process is described by

$$p_\theta(x_{t-1}|x_t) = \mathcal{N}\big(x_{t-1}; \mu_\theta(x_t, t), \Sigma_\theta(x_t, t)\big). \tag{3}$$

The QDM, which integrates QML and DM, is utilized for generative tasks within the quantum domain, including quantum state generation and quantum circuit design. The quantum denoising diffusion model (QDDM)[7] is acknowledged as the leading quantum diffusion method for image generation. It outperforms classical models with similar parameter counts, while leveraging the efficiencies of quantum computing. Fig. 3 shows the framework of QDDM and its image generation process is illustrated in Fig. 2. In our work, we extend the QDDM with a label-guided mechanism to fully leverage the capabilities of QDDM in addressing the QFSL problems. This is achieved by introducing an additional qubit and applying a Pauli-X rotation by an angle of $2\pi y/n$, where $y$ represents the specified label and $n$ denotes the total number of classes.

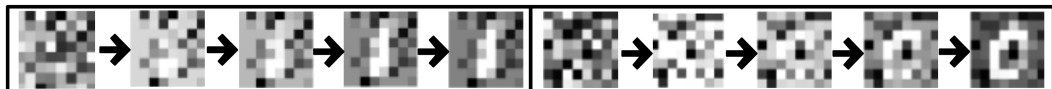

**Figure 2:** Generated images using QDDM under the guidance of different labels. The input to the model is random noise.

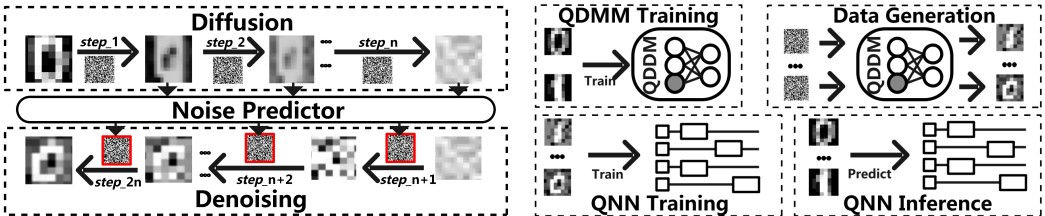

**Figure 3:** Framework of QDDM. **Noise Predictor** is employed to estimate the noise present in the noisy image data.

**Figure 4:** Framework of QDDM-based Label-Guided Generation Inference (QDiff-LGGI). The gray-filled circle represents the embedded label.

## 3  Method

To address the QFSL problems, we propose methods from both data and algorithmic perspectives. From the data perspective, we utilize QDDM to augment the training samples and use the generated data to train QNN, thereby improving the prediction accuracy of QNN on real data. From an algorithmic perspective, we employ two strategies to complete the inference process by guiding QDDM in two distinct stages: diffusion and denoise.

### 3.1  QDiff-Based Label-Guided Generation Inference (QDiff-LGGI)

The size of the training dataset is a critical factor that limits the performance of QNN. The primary reason for the suboptimal performance of QFSL is the limited availability of training data. Thus, from a data perspective, expanding the training dataset can significantly enhance the performance of QFSL. The QDDM is highly effective in generation tasks, making it suitable for augmenting the training dataset. Initially, a small amount of training data is used to train the QDDM. Once trained, the QDDM is employed to expand the training dataset for QNN. This expanded dataset is then used to train the QNN, which in turn improves its inference accuracy on real data.

To enhance the quality of data generated by the QDDM, we employ a label-guided generation method. During the QDDM training process, we perform amplitude encoding on the classical data and angle encoding on the labels. During the data generation process, we use random noise and the label as input, enabling the QDDM to generate data according to the specified label. Fig. 2 illustrates the data generation process under different label guidance. Fig. 4 describes the QDiff-LGGI algorithm.

### 3.2  QDiff-Based Label-Guided Noise Addition Inference (QDiff-LGNAI)

The learning objective of the QDDM outlined in Equation 2 relies on using a noise predictor to estimate the noise in noisy data compared to the actual noise. The noise predictor's estimation is guided by a label, with different labels corresponding to different noise predictions. By using the correct label for guidance, the error between the predicted noise and the actual noise is minimized. Based on this principle, we propose the QDM-Based Label-Guided Noise Addition Inference (Diff-LGNAI) method, shown in the Fig. 5.

We first utilize a small amount of training data to complete the training of the QDDM. Once trained, the noise predictor $\mathcal{P}$ within the QDDM is used for subsequent inference. For a given input $x_0$, the possible labels are $\{L_1, L_2, \ldots, L_m\}$. Noise is gradually added to $x_0$ over $\mathcal{T}$ iterations. Specifically, at each time step $t$, $x_t$ is calculated as $x_{t-1} + \epsilon_t$, where $\epsilon_t \sim \mathcal{N}(x_{t-1}, \mathcal{W}[t])$, and $\mathcal{W}$ represents the noise weight. The noise predictor $\mathcal{P}$ is then employed to estimate the noise in the noisy data $x_t$, guided by various possible labels, resulting in the predicted noise set $\{\mathcal{P}(x_t|L_1), \ldots, \mathcal{P}(x_t|L_m)\}$. We calculate the mean squared error (MSE) between the predicted noise and the actual noise, $\text{MSE}(\mathcal{P}(x_t|L_i), \epsilon_t)$. The error is computed for each possible label, and the label with the minimum

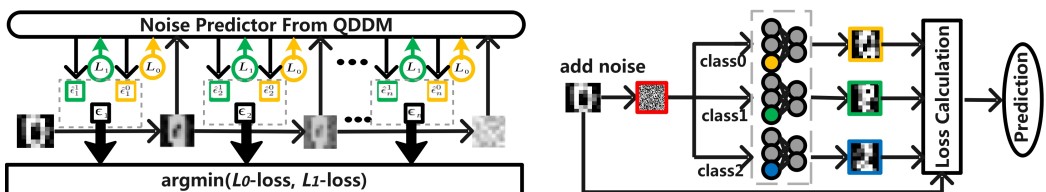

**Figure 5:** Framework of QDDM-based Label-Guided Noise Addition Inference (QDiff-LGNAI). The term $\hat{\epsilon}_m^n$ represents the predicted noise at step $m$ associated with the true noise and the predicted noise under the guidance of different labels $L_i$.

**Figure 6:** Framework of QDDM-based Label-Guided Denoising Inference (QDiff-LGDI). Solid circles in different colors represent distinct embedded labels. The label $n$. $L_0/L_1$-loss denotes the difference between the output images, each framed by a square of varying colors, indicate the generated images guided by different labels $L_i$.

average error over $\mathcal{T}$ iterations is selected as the predicted label:

$$\arg \min_{L_i \in \mathcal{L}} \sum_{t=1}^{\mathcal{T}} \text{MSE}(\mathcal{P}(x_t|L_i), \epsilon_t).$$

### 3.3 QDiff-Based Label-Guided Denoising Inference (QDiff-LGDI)

During the denoising phase of QDDM, the noise predictor is used to estimate the noise present in the noisy data, which is then subtracted from the noisy data. This denoising process is repeated over $\mathcal{T}$ iterations. The noise prediction is guided by labels, with each label producing distinct noise estimates. The data generated under the guidance of the true label is expected to be most similar to the original data. In this framework, we propose the QDiff-Based Label-Guided Denoising Inference (QDiff-LGDI) method.

For an input $x_0$, we gradually add noise to $x_0$ over $\mathcal{T}$ iterations, resulting in progressively noisier data $\{x_1, x_2, \ldots, x_{\mathcal{T}}\}$. Then, we use the noise predictor $\mathcal{P}$ to predict the noise in the noisy data under the guidance of label $L_i$, obtaining $\mathcal{P}(x_{\mathcal{T}}|L_i)$. The predicted noise is subtracted from the noisy data. This denoising process is also performed over $\mathcal{T}$ iterations, producing progressively noise-reduced data $\{x_{\mathcal{T}+1}, x_{\mathcal{T}+2}, \ldots, x_{2\mathcal{T}}\}$, where $x_{\mathcal{T}+t+1}|L_i = x_{\mathcal{T}+t} - \mathcal{P}(x_{\mathcal{T}+t}|L_i)$. We then use the MSE loss to calculate the error between the generated data and the noisy data under the guidance of different labels $L_i$, and the predicted label is chosen such that

$$\arg \min_{L_i \in \mathcal{L}} \sum_{t=0}^{\mathcal{T}} \text{MSE}(x_t, x_{2\mathcal{T}-t}|L_i).$$

## 4 Experiment

In this section, we first outline the fundamental settings of our experiment. We then design a series of experiments to explore the following specific questions, each addressed in a dedicated subsection:

- What are the performance advantages of our proposed three QDiff-based algorithms compared to other baseline methods?
- What factors influence the performance of our algorithms?
- How effectively does our algorithms solve the zero-shot problem?

### 4.1 Basic Experimental Settings

In this section, we provide a detailed description of the dataset used for the experiments, the baseline algorithms, and the parameter settings of the algorithms.

**Dataset.** During the experiment, we use the Digits MNIST[18], MNIST[19], and Fashion MNIST[20] datasets. For the 2-way $k$-shot tasks, we select classes 0 and 1 from both the Digits MNIST and MNIST datasets, and the T-shirt and Trouser classes from the Fashion MNIST dataset. For the 3-way $k$-shot tasks, we choose classes 0, 1, and 2 from both the Digits MNIST and MNIST datasets, and the T-shirt, Trouser, and Pullover classes from the Fashion MNIST dataset. During training, for the

**Table 1:** Performance comparison of QDiff-based algorithms across various tasks, with $\mathcal{T} = 5$. Each algorithm is evaluated using 5 random seeds to report mean performance and standard error. The best-performing algorithm for each task is highlighted in blue.

| Dataset | Tasks | LGDI | LGNAI | LGGI | QMLP | C14 | OPTIC | Quantumnat |
|---|---|---|---|---|---|---|---|---|
| Digits | 2w-01s | $0.975_{\pm0.059}$ | $0.978_{\pm0.003}$ | $0.992_{\pm0.009}$ | $0.764_{\pm0.108}$ | $0.505_{\pm0.175}$ | $0.525_{\pm0.133}$ | $0.751_{\pm0.147}$ |
| | 2w-10s | $0.983_{\pm0.006}$ | $0.997_{\pm0.002}$ | $0.984_{\pm0.012}$ | $0.892_{\pm0.086}$ | $0.627_{\pm0.086}$ | $0.886_{\pm0.193}$ | $0.722_{\pm0.186}$ |
| | 3w-01s | $0.525_{\pm0.001}$ | $0.635_{\pm0.007}$ | $0.573_{\pm0.069}$ | $0.338_{\pm0.087}$ | $0.447_{\pm0.193}$ | $0.475_{\pm0.021}$ | $0.555_{\pm0.013}$ |
| | 3w-10s | $0.857_{\pm0.015}$ | $0.801_{\pm0.008}$ | $0.632_{\pm0.035}$ | $0.355_{\pm0.059}$ | $0.481_{\pm0.183}$ | $0.698_{\pm0.121}$ | $0.687_{\pm0.156}$ |
| MNIST | 2w-01s | $0.943_{\pm0.002}$ | $0.965_{\pm0.003}$ | $0.805_{\pm0.093}$ | $0.675_{\pm0.067}$ | $0.567_{\pm0.064}$ | $0.845_{\pm0.149}$ | $0.701_{\pm0.162}$ |
| | 2w-10s | $0.953_{\pm0.011}$ | $0.978_{\pm0.005}$ | $0.915_{\pm0.079}$ | $0.817_{\pm0.048}$ | $0.810_{\pm0.152}$ | $0.807_{\pm0.173}$ | $0.727_{\pm0.151}$ |
| | 3w-01s | $0.475_{\pm0.003}$ | $0.505_{\pm0.007}$ | $0.428_{\pm0.035}$ | $0.325_{\pm0.027}$ | $0.503_{\pm0.122}$ | $0.477_{\pm0.159}$ | $0.501_{\pm0.012}$ |
| | 3w-10s | $0.720_{\pm0.016}$ | $0.825_{\pm0.008}$ | $0.405_{\pm0.022}$ | $0.547_{\pm0.085}$ | $0.607_{\pm0.142}$ | $0.770_{\pm0.191}$ | $0.527_{\pm0.078}$ |
| Fashion | 2w-01s | $0.738_{\pm0.007}$ | $0.768_{\pm0.007}$ | $0.898_{\pm0.036}$ | $0.688_{\pm0.064}$ | $0.581_{\pm0.187}$ | $0.765_{\pm0.149}$ | $0.583_{\pm0.181}$ |
| | 2w-10s | $0.755_{\pm0.020}$ | $0.805_{\pm0.002}$ | $0.895_{\pm0.066}$ | $0.731_{\pm0.035}$ | $0.773_{\pm0.099}$ | $0.793_{\pm0.157}$ | $0.887_{\pm0.129}$ |
| | 3w-01s | $0.453_{\pm0.008}$ | $0.433_{\pm0.001}$ | $0.483_{\pm0.012}$ | $0.331_{\pm0.098}$ | $0.332_{\pm0.172}$ | $0.473_{\pm0.128}$ | $0.622_{\pm0.063}$ |
| | 3w-10s | $0.655_{\pm0.018}$ | $0.735_{\pm0.004}$ | $0.585_{\pm0.025}$ | $0.647_{\pm0.015}$ | $0.527_{\pm0.173}$ | $0.593_{\pm0.139}$ | $0.653_{\pm0.032}$ |
| Average | | $0.754_{\pm0.015}$ | $0.795_{\pm0.004}$ | $0.719_{\pm0.045}$ | $0.574_{\pm0.060}$ | $0.546_{\pm0.140}$ | $0.678_{\pm0.150}$ | $0.666_{\pm0.120}$ |

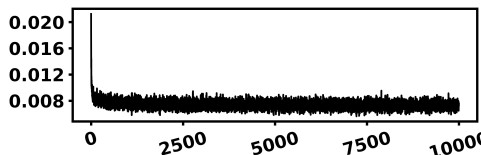

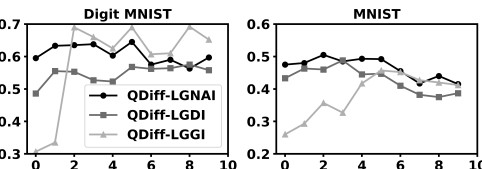

**Figure 7:** Training Loss Trends during QDDM Model Training.

**Figure 8:** Performance of QDiff-based algorithms on 3-way, 1-shot task under varying diffusion and denoising step configurations.

one-shot task, we select one image from each category, and for the ten-shot task, we select ten images from each category. In the inference phase, we use 200 images from each category to construct the evaluation dataset.

**Baselines and Parameters Setting.** For the selection of baselines, we choose four representative QNN structures in the current QML domain to accomplish the QFSL task [9–12]. The frameworks of the four QNNs are shown in Fig. 1. During the training of the QNN, we resize the image data to $8 \times 8$ and utilize amplitude encoding to convert classical data into quantum states. Adam optimizer is employed with a learning rate of $0.001$ and cross entropy loss is minimized over 40 iterations.

**QDDM Training.** Before applying QDiff-based algorithms to finish the QFSL task, it is essential to obtain a well-trained QDDM model. For training the QDDM, we utilize a label-guided quantum dense architecture, where the label is embedded using an $RX$ rotation, and the strongly entangling layers[21] are used to transform the data. The training process of QDDM involves using the Adam optimizer with $10,000$ iterations. The model architecture and learning rate are tailored to each dataset. For the Digits MNIST dataset, the circuit consists of 47 layers with a learning rate of $0.00097$. For the MNIST dataset, it comprises 60 layers with a learning rate of $0.00211$, and for the Fashion MNIST dataset, the circuit includes 121 layers with a learning rate of $0.00014$.

### 4.2 Performance Analysis of QDiff-based QFSL Algorithms

During the QDDM training phase, in the $n$-way, $k$-shot setting, $k$ images are selected from each of the $n$ categories, resulting in a total of $n \times k$ images. Fig. 7 illustrates the trend of training loss while training QDDM on Digits MNIST dataset. As training progresses, the decreasing training loss reflects the improved accuracy of the noise predictor in estimating noise, resulting in denoised images that closely resemble the target images.

Table 1 presents the performance of the QDiff-based QFSL algorithm compared to other baselines for 2-way 1-shot, 2-way 10-shot, 3-way 1-shot, and 3-way 10-shot scenarios. The results in the table demonstrate that the QDiff-based algorithm achieves state-of-the-art performance. We also assess the performance of the QDiff-based algorithms on a 3-way, 1-shot task using the Digits MNIST dataset on a real quantum computer (IBM_Almaden). The results, as shown in Fig. 9, reveal a slight performance decline due to noise inherent in the quantum hardware. Nevertheless, the decrease is marginal, indicating that our algorithms perform robustly even in noisy processors.

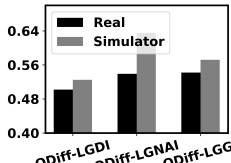 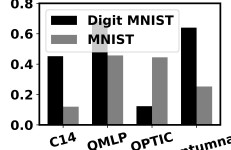 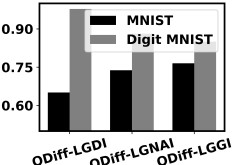 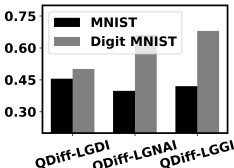

**Figure 9:** Performance of QDiff-based algorithms for the 3-way, 1-shot task in (IBM_Almaden).

**Figure 10:** Performance of QDiff-LGGI on the 3-way, 1-shot task across different QNNs.

**Figure 11:** Performance of QDiff-based algorithms on the zero-shot, two-class classification task.

**Figure 12:** Performance of QDiff-based algorithms on the zero-shot, three-class classification task.

### 4.3 Factors Impacting the Effectiveness of QDiff-based QFSL Algorithms

In this section, we explore the factors that influence the performance of QDiff-based algorithms, including the impact of diffusion and denoising steps, the quantity of training data, and the selection of QNNs utilized in QDiff-LGGI.

With variations in the number of diffusion and denoising steps, the performance of QDiff-based algorithms on the Digits MNIST and MNIST datasets varies, as shown in Fig. 8. The experimental results demonstrate that QDiff-LGGI is highly sensitive to the number of diffusion and denoising steps. As the number of steps increases, QDiff-LGGI is improved, indicating that more steps result in the generation of higher-quality images that are closer to the target data domain. However, an excessive number of steps may cause the original image to degrade too much into noise during the diffusion stage. Consequently, during the denoising stage, the reconstruction process may overemphasize the label, resulting in a mismatch with the original image. This mismatch negatively impacts inference performance, and the phenomenon is more pronounced in QDiff-LGNAI and QDiff-LGDI.

The quantity of training data used to train the QDDM significantly influences the performance of the QDiff-based QFSL algorithm. We compare the performance of the QDDM when trained with one-shot versus ten-shot learning. Table 1 presents the performance comparison across different datasets. The results indicate that increasing the amount of training data enhances the training of the QDDM, which subsequently leads to improved performance of the QDiff-based algorithms when the QDDM is well-trained.

QDiff-LGGI uses generated images to train QNN, which is then used for inference. The performance of inference varies depending on the QNN architecture. Fig. 10 shows that different QNNs produce varying inference results, likely due to differences in the quantum circuits' expressibity and entangling capabilities[10].

### 4.4 Zero-Shot Learning with QDiff-based QFSL Algorithms

We evaluate the effectiveness of our methods in solving zero-shot tasks. The QDDM model is initially trained on the MNIST dataset and then applied within QDiff-based algorithms for evaluation on the Digits MNIST dataset. Conversely, we also train the QDDM model on the Digits MNIST dataset and assess its performance on the MNIST dataset. We evaluate performance on both 2-way and 3-way zero-shot classification tasks. The results of these experiments are shown in Figs. 11 and 12. Based on these results, we conclude that QDiff-based algorithms demonstrate strong performance in zero-shot scenarios when the training and evaluation datasets belong to similar domains.

## 5 Conclusion and Future Work

In this work, we introduce quantum diffusion model (QDM) to tackle the challenges of quantum few-shot learning. We propose three algorithms—QDiff-LGDI, QDiff-LGNAI, and QDiff-LGGI—developed from both data-driven and algorithmic perspectives. These algorithms demonstrate significant performance improvements over existing baselines. Nevertheless, the current limitations of the QDM confine its applicability to relatively simple datasets. Future research could focus on enhancing the QDM's capability and expanding its application to other QML tasks, such as quantum object detection and quantum semantic segmentation.

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
