# OpenReview forum: "Quantum Diffusion Models for Few-Shot Learning"
_NeurIPS.cc/2024/Workshop/MLNCP — Submitted to MLNCP_

### Official Review · Reviewer_pY4k · 2024-10-03

**Rating:** 5
**Confidence:** 3

**Review:**

This paper proposed three approaches to perform few shot learning with quantum denoise diffusion model (QDDM). Specifically, (1) LGGI: train a QDDM with few shot samples and generate synthetic samples to augment the dataset. Then, train a quantum NN (QNN) for classification; (2) LGNAI: use the trained noise predictor network to predict the labels by matching the predicted noise with the actual noise; (3) LGDI: denoise the image with trained noise predictor and extract the label by measuring the difference of denoised image w.r.t. true image. Empirically, the authors demonstrates the improved performance compared to directly train QNN with few shot samples.

These three methods are intuitive but I have several main concerns.
1. It seems a bit counter-intuitive that you can train a generative model with few-shot samples and the augmented dataset allows one to get better performance. Typically, generative model is much harder to train (more informative) than simple classifier. This seems like a detour to what we want to achieve. However, there is no explanation on why the proposed approach performs better.

2. The other two approaches (LGNAI and LGDI) are intuitive but also lacks the explanation on why they are better.  Since the dataset are pretty small and tasks are pretty simple, I am not sure this will hold in more complicated tasks.

3. I wouldn't say the performance drop in actual quantum computer is marginal, it seems like a noticeable drop compared to simulation.

---

### Official Review · Reviewer_L7U4 · 2024-10-07
**Some results on QML and diffusion models for few-shot learning**

**Rating:** 5
**Confidence:** 1

**Review:**

This is a difficult paper to evaluate.  It is motivated by the desire for few-shot learning, which is an open field with no established answer yet, and diffusion models are one possible solution.

The authors want to explore whether "quantum machine learning" (QML) is suited to this task.  They studied three approaches, called LGGI, LGNAI, and LGDI, and evaluated them on three test problems.  I found it difficult to understand the impact of this result, admittedly in part because I do not work in QML, and having previous experience in a quantum information group have grown somewhat jaded and cynical about the whole field.

But it looks like they’re applying the right methodology, and it's mathematically rigorous, so this might make the cutoff depending on the number of papers received.

Estimate: 20-40th percentile.

---

### Official Review · Reviewer_tyAo · 2024-10-07

**Rating:** 5
**Confidence:** 3

**Review:**

**Summary:** \
The article introduces a framework based on the Quantum Denoising Diffusion Model (QDDM) [1], proposing three methods to address the few-shot learning (FSL) problem. One method adopts a data-centric approach, while the other two are algorithmically focused. The numerical experiments conducted on paradigmatic datasets such as MNIST and its variants, both on simulators and real quantum devices, demonstrate the robustness of the proposed framework.

\
\
**Main Review:**
\
**Writing:** The majority of the article is well-written; however, some key details are missing, which would enhance the clarity for the reader. For example, the authors do not provide the specifications of the quantum neural network used for inference in their QDiff-LGGI method. Additionally, the number of qubits used for QDDM training is not explicitly stated. While it is understood that the number would likely be O(log n), where n represents the total number of image pixels, explicitly mentioning this in the text would improve readability and comprehension.
\
\
**Comparison to classical methods:** A significant drawback of the article is the lack of comparison with classical methods. Specifically, it is unclear how the performance would change if the quantum components were removed from the proposed framework. It is not entirely clear that there is an advantage of using "quantumness" of quantum computing in this particular setting.
\
\
**Quantum dense architecture:** During the training of QDDM, as the quantum dense architecture is being utilized, I am concerned about the trainability of this quantum circuit as quantum neural networks are prone to the phenomenon of barren plateuas[2]. The authors failed to address this question in the article and if addressed can only make the paper stronger.
\
The use of a quantum dense architecture during QDDM training raises concerns about the trainability of the quantum circuit, particularly given the well-documented issue of barren plateaus in quantum neural networks [2]. The authors do not address this concern in the article. Including a discussion on this aspect would strengthen the paper. Additionally, it is recommended that the authors justify the necessity of a strongly entangling layer within the quantum circuit, as some studies (e.g., [3]) suggest that removing entanglement from quantum models can yield comparable or even improved performance, implying that entanglement may not be essential for classification tasks.
\
\
**Scaling and computational cost:** For the QDiff-LGNAI and QDiff-LGDI methods, the authors should provide more information on the scalability and computational cost of these approaches. Given that the noise predictor $\mathcal{P}$ must be utilized at each denoising step to compute $\mathcal{P}(x_t | L_i)$, it is important to understand the computational implications and scaling behavior of these methods as the number of classes in the dataset increases.
\
\
**Error Bars:** The absence of error bars in Figures 8-12 is a notable omission. Error bars are crucial for understanding the variability and reliability of the results. The authors should provide error bars to give a clearer indication of the robustness of their findings.
\
\
**Code:** Another area for improvement is the lack of anonymized source code, which limits the ability of other researchers to reproduce and extend the work. Providing accessible code would significantly enhance the transparency and impact of the study.
\
\
\
[1] Kölle, Michael, et al. "Quantum Denoising Diffusion Models." arXiv preprint arXiv:2401.07049 (2024). \
[2] McClean, Jarrod R., et al. "Barren plateaus in quantum neural network training landscapes." Nature communications 9.1 (2018): 4812. \
[3] Bowles, Joseph, et al. "Better than classical? the subtle art of benchmarking quantum machine learning models." arXiv preprint arXiv:2403.07059 (2024).

---

### Decision · Program_Chairs · 2024-10-10

Reject